# Vision Mamba Mender

**Jiacong Hu**[1,3]**, Anda Cao**[1]**, Zunlei Feng**[2,3,4*]**,**
**Shengxuming Zhang**[2]**, Yi Wang**[1]**, Lingxiang Jia**[1]**, Mingli Song**[1,3,4]

[1]College of Computer Science and Technology, Zhejiang University,
[2]School of Software Technology, Zhejiang University,
[3]State Key Laboratory of Blockchain and Data Security, Zhejiang University,
[4]Hangzhou High-Tech Zone (Binjiang) Institute of Blockchain and Data Security

{jiaconghu,caoanda,zunleifeng}@zju.edu.cn,
{zsxm1998,y_w,lingxiangjia,brooksong}@zju.edu.cn

## Abstract

Mamba, a state-space model with selective mechanisms and hardware-aware architecture, has demonstrated outstanding performance in long sequence modeling tasks, particularly garnering widespread exploration and application in the field of computer vision. While existing works have mixed opinions of its application in visual tasks, the exploration of its internal workings and the optimization of its performance remain urgent and worthy research questions given its status as a novel model. Existing optimizations of the Mamba model, especially when applied in the visual domain, have primarily relied on predefined methods such as improving scanning mechanisms or integrating other architectures, often requiring strong priors and extensive trial and error. In contrast to these approaches, this paper proposes the Vision Mamba Mender, a systematic approach for understanding the workings of Mamba, identifying flaws within, and subsequently optimizing model performance. Specifically, we present methods for predictive correlation analysis of Mamba's hidden states from both internal and external perspectives, along with corresponding definitions of correlation scores, aimed at understanding the workings of Mamba in visual recognition tasks and identifying flaws therein. Additionally, tailored repair methods are proposed for identified external and internal state flaws to eliminate them and optimize model performance. Extensive experiments validate the efficacy of the proposed methods on prevalent Mamba architectures, significantly enhancing Mamba's performance. For more information, please visit https://vision-mamba-mender.github.io/.

## 1 Introduction

Deep learning has demonstrated outstanding performance in various fields of artificial intelligence, with deep neural networks such as convolutional neural networks (CNNs) [1, 2] and Vision Transformer [3, 4] dominating the field of computer vision. However, the limited receptive field [5, 6] of CNNs and the quadratic computational complexity [7, 8] of Transformers constrain their further development. To overcome these limitations, an increasing number of studies have attempted to propose more advanced models [9–12]. Recently, Mamba [13], based on the state space model [11], has become the focus of these research efforts. Mamba can maintain nearly linear computational complexity while achieving a global receptive field, leading to outstanding performance. Consequently, it has been widely adopted in the field of computer vision [14–16].

---

*Corresponding author.

38th Conference on Neural Information Processing Systems (NeurIPS 2024).

To enhance the performance of Mamba in visual tasks, existing works mainly focus on improving the architecture of Mamba [17–21]. For instance, Zhu et al.[17] proposed Vision Mamba, which adds branches to the original Mamba to simultaneously process image sequences in both forward and backward directions. Nearly simultaneously, Liu et al.[18] introduced VMamba, which adopts a four-way scanning strategy on top of the original Mamba to achieve a more comprehensive global receptive field. Other improvements include PlainMamba [19], Mamba-ND [20], SiMBA [21], and MambaMixer [22], all of which are variants of the original Mamba framework.

However, the aforementioned methods that optimize the Mamba model by improving its architecture are predefined and require strong prior knowledge and extensive trial and error. Additionally, Yu et al. [23] recently pointed out in their latest research that the current improvements made to Mamba-based visual models are unnecessary for visual tasks, especially for visual recognition tasks. This opposing view underscores the necessity and urgency of further optimizing Mamba models in the field of computer vision. Hence, unlike the aforementioned pre-optimization methods, this paper attempts to analyze the working mechanism of Mamba from a post-perspective, identify the reasons of flaws leading to incorrect prediction results, and automatically rectify them to further enhance the performance of Mamba models. Moreover, this approach is applicable to all Mamba-like models.

Based on this idea, in this paper, we propose Vision Mamba Mender, a systematic approach to understanding the working mechanism of Mamba from a post-perspective, identifying flaws within it, and rectifying them to ultimately improve model performance. In understanding the operational mechanism of the Mamba model, we categorize the computational process of Mamba into external state interaction and internal state interaction.

Along these two perspectives of external and internal states, we introduce a state correlation analysis method tailored for Mamba to establish the correlation between hidden states and predicted results. Additionally, we define external state correlation scores and internal state correlation scores to quantitatively analyze differences in state correlations, revealing flaws existing respectively in the external and internal states. Specifically, external state flaws refer to instances where correct model predictions in certain states are predominantly associated with foreground regions, while incorrect predictions are primarily linked to background regions. Internal state flaws, on the other hand, pertain to cases where correct predictions within a class are correlated with the same regions within the state and exhibit low overall complexity, whereas incorrect predictions within the class focus on different internal regions of the state and demonstrate higher overall complexity.

Furthermore, we propose repair methods tailored for addressing both external state flaws and internal state flaws. Specifically, in the repair of external state flaws, we impose constraints on the external state correlations within certain modules of Mamba to increase their correlation with foreground information during prediction. On the other hand, in the repair of internal state flaws, we impose constraints on the internal state correlations within certain modules of Mamba to enhance their correlation with genuine class-specific internal information during prediction. Through extensive experimentation, we demonstrate that the proposed Vision Mamba Mender is applicable to state-of-the-art Vision Mamba architectures.

The contributions of this paper are summarized as follows:

- We propose a novel post-hoc optimization method named Vision Mamba Mender. This method is applicable to existing state-of-the-art Vision Mamba architectures, identifying and repairing flaws in the Mamba model's mechanisms for visual recognition tasks from a post-hoc perspective, ultimately enhancing the model's performance.
- We introduce a state correlation analysis method and correlation score definitions for Mamba from both external and internal hidden state perspectives. These methods are used to identify flaws. Additionally, we introduce a state correlation constraint method to rectify these flaws.
- Extensive experiments demonstrate that the proposed Vision Mamba Mender can effectively identify and repair flaws in the Mamba model without introducing additional parameters, significantly improving the performance of the Vision Mamba.

## 2   Preliminaries

Mamba [13] is a novel and efficient sequence modeling model composed of multiple uniformly stacked Mamba blocks. Each Mamba block is constructed based on a selective state space model

(SSM). Unlike the previous time-invariant SSM [11], it allows the parameters of the SSM to depend on the input, enhancing the model's expressive capacity. Furthermore, inspired by H3 [24] and Gated MLP [25], each Mamba block also incorporates modules such as causal convolution and gating mechanisms.

As illustrated in Figure 1, given the hidden state $h_n^{(\ell)-1}$ of the $i$-th token in the input of the $\ell$-th Mamba block, the computation of the hidden state $h_n^{(\ell)}$ of the $i$-th token in the output of the $\ell$-th Mamba block is as follows:

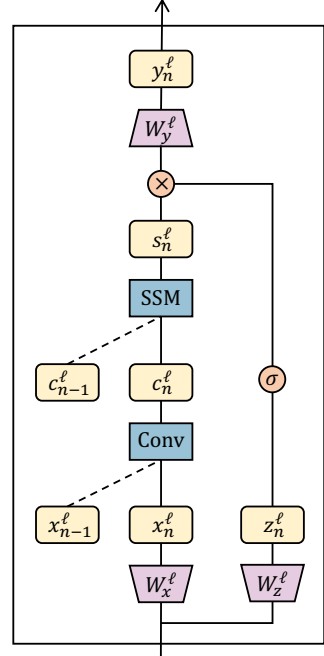

$$x_n^{(\ell)} = \text{SiLU}(h_n^{(\ell)-1} \cdot W_x^{(\ell)}), \tag{1}$$

$$c_1^{(\ell)}, c_2^{(\ell)}, \ldots, c_n^{(\ell)} = \text{cacusal-Conv1D}(x_1^{(\ell)}, x_2^{(\ell)}, \ldots, x_n^{(\ell)}), \tag{2}$$

$$s_n^{(\ell)} = \text{selective-SSM}(c_1^{(\ell)}, c_2^{(\ell)}, \ldots, c_n^{(\ell)}), \tag{3}$$

$$z_n^{(\ell)} = \text{SiLU}(h_n^{(\ell)-1} \cdot W_z^{(\ell)}), \tag{4}$$

$$y_n^{(\ell)} = (s_n^{(\ell)} \odot z_n^{(\ell)}) \cdot W_y^{(\ell)}, \tag{5}$$

where SiLU(.), causal-Conv1D(.), and selective-SSM(.) denote the activation function, the casual 1D convolution, and the selective state model, respectively. $W_x^{(\ell)}$, $W_z^{(\ell)}$, and $W_y^{(\ell)}$ are the projection matrices for the linear operations in the Mamba block, while $x$, $z$, $c$, $s$, and $y$ represent the intermediate states within the Mamba block.

Finally, by applying a residual connection, the hidden state $h_n^{(\ell)}$ is obtained as follows:

$$h_n^{(\ell)} = h_n^{(\ell)-1} + y_n^{(\ell)}. \tag{6}$$

In Eqn. (5), $\odot$ denotes the Hadamard product. The term $z_n^{(\ell)}$ in Eqn. (4) is computed through a separate pathway, serving as a gating mechanism to regulate the information flow in the main pathway.

Figure 1: The computational process of a Mamba block.

However, the original Mamba block is designed for one-dimensional sequences and is not suitable for handling multidimensional visual data, particularly for vision tasks requiring spatial awareness. Existing vision Mamba architectures enhance the basic Mamba block to accommodate these requirements, such as ViM [17], VMamba [18], PMamba [19], Mamba-ND [20], and SiMBA [21]. Unlike the predefined optimizations, our approach analyzes the working mechanism of the Mamba model post-hoc, identifying flaws and making repairs to further enhance the model's performance.

## 3 Where Do Flaws Occur?

Identifying flaws in Mamba first requires an understanding of how Mamba operates. Some studies have provided empirical evidence to elucidate the mechanisms of Mamba models in the NLP domain, such as their contextual learning ability [26], factual recall capability [27], and interpretability [28]. However, elucidating the operational mechanisms of Mamba models in the visual domain remains a significant challenge. Ali et al. [29] established a connection between the selective SSM within the Mamba block and the self-attention mechanism in Transformers, allowing the selective SSM to represent the interaction process between any two states by constructing a self-attention matrix, which is utilized for image feature attribution. However, focusing solely on the selective SSM within the Mamba block is far from sufficient, as causal convolution, as shown in Eqn. (2), also participates in the interaction between states. Moreover, other computational modules within the Mamba block used for state interaction must also be considered.

In this section, we first investigate the working mechanisms of Mamba. We summarize the computational processes within Mamba as state interactions[1]. These interactions are categorized into *external state interactions* (where a state interacts with other states to form a new state, as shown in Eqn. (2) and (3) and *internal state interactions* (where a state interacts only with its internal information to form a new state, as shown in Eqn. (1), (4), and (5). We explore the operational mechanisms of Mamba from both the external and internal state interaction perspectives to identify flaws.

---

[1]In Mamba models used for visual recognition, each state corresponds to an image patch or a learnable classification token.

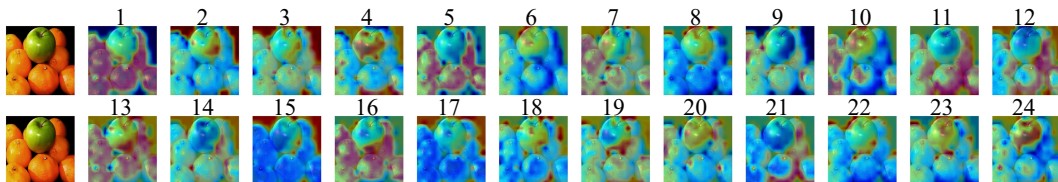

Figure 2: Visualization of the external state correlation $\mathbf{e}^{(\ell,s)}$ of the output $s_n^{(\ell)}$ from the selective-SSM module in different ViM [17] blocks. The depth of the ViM blocks increases from left to right.

## 3.1 External State Correlation Analysis

To understand the impact of external states on predictions within Mamba's working mechanisms, we propose the Grad-ESC method, inspired by Grad-CAM [30], which uses gradient and activation information to assess the importance of each neuron in decision-making. Grad-ESC calculates the correlation between external states and the model's prediction outcomes, termed external state correlation. Unlike the attention mechanism derived by Ali et al. [29], which applies only to the selective SSM module within the Mamba block, Grad-ESC allows for correlation analysis of the outputs from any module within the Mamba block.

Specifically, given the predictive distribution $p$ output by the Mamba model and the true class $k$ of the input sample, the calculation process for external state correlation $\mathbf{e}^{(\ell,s)} \in \mathbb{R}^{H \times W}$ (where $H$ and $W$ represent the height and width of the input image) of the states $\{s_n^{(\ell)}\}_{n=1}^{N}$ output by the selective SSM module (where $s_n^{(\ell)} \in \mathbb{R}^D$, $N$ and $D$ are the number and dimensionality of the states) is as follows:

$$\mathbf{e}^{(\ell,s)} = \mathcal{R}(\bar{s}_1^{(\ell)}, \bar{s}_2^{(\ell)}, \ldots, \bar{s}_N^{(\ell)}), \bar{s}_n^{(\ell)} = \mathbb{E}_D(g^{(\ell,s)} \odot s_n^{(\ell)}), g^{(\ell,s)} = \frac{1}{N} \sum_{n=1}^{N} \frac{\partial p^k}{\partial s_n^{(\ell)}}, \quad (7)$$

where $g^{(\ell,s)} \in \mathbb{R}^D$ represents the weights used to weight the state $s_n^{(\ell)}$ along its dimensions based on gradient information, $\mathbb{E}_D(\cdot)$ denotes the average across all dimensions, $\hat{s}_n^{(\ell)} \in \mathbb{R}$ signifies the degree of correlation between the $n$-th state and the prediction after integrating gradient information, and $\mathcal{R}$ denotes the operation of reshaping and scaling to the original image size after retaining only the states corresponding to the image patches.

As illustrated in Figure 2, taking the state $s_n^{(\ell)}$ as an example, we visualize the external state correlations of $s_n^{(\ell)}$ in different blocks of ViM [17]. It can be observed that the external state correlations of $s_n^{(\ell)}$ vary across different blocks; some blocks exhibit correlations with foreground regions (such as the 1st layer), while others correlate with background regions (such as the 2nd layer). We consider correlations with the foreground to be interpretable, whereas correlations with the background are deemed uninterpretable. To quantify the interpretability of external state correlations, we propose the following definition of an external state correlation score based on commonly used interpretable evaluation methods such as perturbation test [29, 31] and segmentation test [32, 30, 33]:

**Definition 1 (External Correlation Score).** *Given a pre-trained Mamba model $\mathcal{F}(\cdot)$, an input image $i$, foreground annotations $m$ of the input image, and external state correlation $\mathbf{e}^{(\ell,s)}$ computed through the proposed method, the external correlation score is defined as follows:*

$$\text{ECS}(\mathbf{e}^{(\ell,s)}) = \underbrace{\frac{\text{softmax}(\mathcal{F}(\mathbf{e}^{(\ell,s)+} \odot i))}{\text{softmax}(\mathcal{F}(\mathbf{e}^{(\ell,s)-} \odot i))}}_{\text{perturbation test}} \times \underbrace{\text{IoU}(\mathbf{e}^{(\ell,s)+}, m)}_{\text{segmentation test}}, \quad (8)$$

where $\mathbf{e}^{(\ell,s)+} = \mathbb{1}(\mathbf{e}^{(\ell,s)} \geq \alpha)$ denotes the retention of important regions (those greater than the threshold $\alpha$) in $\mathbf{e}^{(\ell,s)}$, setting them to 1, and the removal of unimportant regions (those less than the threshold $\alpha$), setting them to 0, for use in negative perturbation tests. Conversely, $\mathbf{e}^{(\ell,s)-} = \mathbb{1}(\mathbf{e}^{(\ell,s)} < \alpha)$ denotes the removal of important regions and the retention of unimportant regions, for use in positive perturbation tests. Additionally, $\text{ECS}(\mathbf{e}^{(\ell,s)}) \in [0, +\infty)$, where a higher value indicates a higher correlation score.

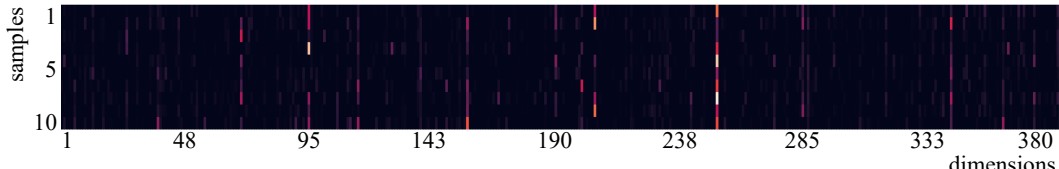

Figure 3: Visualization of the internal state correlations $\mathbf{i}_n^{(\ell,x)}$ of the output states $s_n^{(\ell)}$ from the linear mapping module $W_s^{(\ell)}$ in ViM [17] for samples of the same class. The horizontal axis represents the state dimensions, and the vertical axis represents the samples.

## 3.2 Internal State Correlation Analysis

To comprehend the influence of internal states on prediction outcomes within the operational mechanism of Mamba, we introduce the Grad-ISC method. This method is utilized for computing the degree of correlation between the internal states and model predictions, termed as the state-internal correlation.

Specifically, given the predicted distribution $p$ outputted by the Mamba model and the true class $k$ of the input sample, let's take the example of the $n$-th state $x_n^{(\ell)} \in \mathbb{R}^D$, which is the output of the linear mapping matrix $W_x^{(\ell)}$. The computation process for the corresponding internal state corelation $\mathbf{i}_n^{(\ell,x)} \in \mathbb{R}^D$ is as follows:

$$\mathbf{i}_n^{(\ell,x)} = g_n^{(\ell,x)} \odot x_n^{(\ell)}, g_n^{(\ell,x)} = \frac{\partial p^k}{\partial x_n^{(\ell)}}, \tag{9}$$

where $g_n^{(\ell,x)} \in \mathbb{R}^D$ denotes the weights applied to the dimensions of state $x_n^{(\ell)}$ using gradient information. Similarly, the Grad-ISC method can perform internal state correlation analysis on the output of any module within the Mamba block.

Continuing with the example of state $x_n^{(\ell)}$, we visualize the internal state correlations $\mathbf{i}_n^{(\ell,x)}$ for samples belonging to the same class in ViM [17], as shown in Figure 3. It can be observed that for the same class, the regions of internal state correlation are relatively consistent. We posit that the more consistent and simpler the internal state correlation regions are for samples of the same class, the more interpretable they are. Conversely, the more inconsistent and complex the internal state correlation regions are for samples of the same class, the more difficult they are to interpret. To quantify the interpretability of internal state correlations, we propose a novel definition for the Internal Correlation Score:

**Definition 2 (Internal Correlation Score).** *Given J samples belonging to the same class and the internal state correlation* $\mathbf{i}_n^{(\ell,x)} \in \mathbb{R}^D$ *computed using the proposed method for a particular sample, the Internal Correlation Score is defined as follows:*

$$\text{ICS}(\mathbf{i}_n^{(\ell,x)}) = \underbrace{\mathbb{E}_D\big(\frac{1}{j}\sum_{j=1}^J \mathbf{i}_{n,j}^{(\ell,x)+}\big)}_{\text{simplicity}} \times \underbrace{\mathbb{E}_D\big(\frac{\frac{1}{J}\sum_{j=1}^J \mathbf{i}_{n,j}^{(\ell,x)+}}{\mathbf{i}_{n,1}^{(\ell,x)+} \oplus \mathbf{i}_{n,2}^{(\ell,x)+} \oplus \cdots \oplus \mathbf{i}_{n,J}^{(\ell,x)+}}\big)}_{\text{homogeneity}}, \tag{10}$$

where $\mathbf{i}_{n,j}^{(\ell,x)}$ represents the internal state correlation for the $j$-th sample, $\mathbf{i}_{n,j}^{(\ell,x)+} = \mathbb{1}(\mathbf{i}_{n,j}^{(\ell,x)} \geq \beta)$ denotes the binarized internal correlation (set to 1 if greater than $\beta$, otherwise set to 0), $\oplus$ denotes the XOR operation, and $\mathbb{E}_D(\cdot)$ denotes the average over all dimensions. Furthermore, $\text{ICS}(\mathbf{i}^{(\ell,x)}) \in [0, +\infty)$, where a higher value indicates a higher correlation score.

## 3.3 Identifying Flaw through Correlation Analysis

To analyze flaws within the Mamba model, we examined the external and internal state correlation scores under different conditions using the ViM model and the ImageNet-10 dataset. This analysis allowed us to observe variations in correlation relationships across different states.

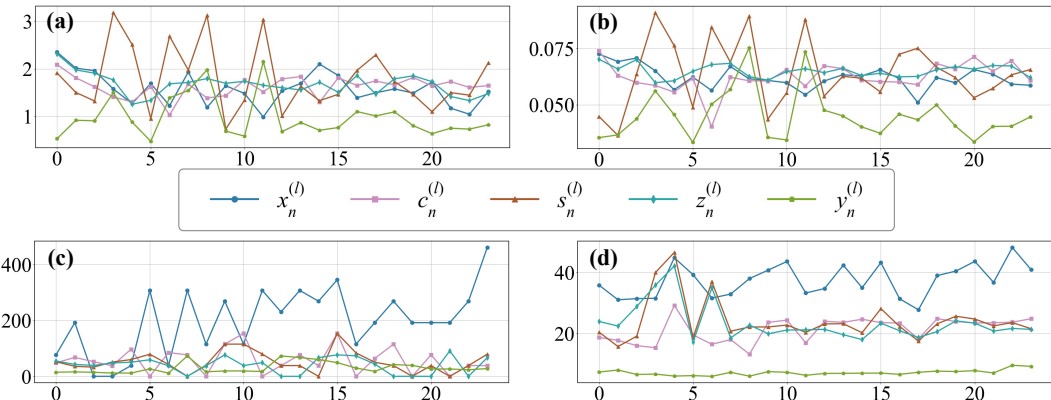

Figure 4: Comparison of external and internal state correlation scores across different blocks in the Mamba model between simple and difficult samples. **(a)** and **(b)** show the external state correlation scores for simple and difficult samples, respectively. **(c)** and **(d)** present the internal state correlation scores for simple and difficult samples, respectively.

**Flaws in External State Correlation.**  To uncover flaws in the external correlations of states, we compare the scores of external state correlations in the Mamba model between simple and difficult samples, as illustrated in Fig. 4(a) and (b). For simple samples, it can be observed that across different blocks, the external correlation scores of states $x_n^{(\ell)}$, $c_n^{(\ell)}$, $s_n^{(\ell)}$, and $z_n^{(\ell)}$ are all better than those of state $y_n^{(\ell)}$, especially in deeper blocks. Furthermore, by comparing simple samples (a) with difficult samples (b), it can be noted that the external correlation scores of all states in all blocks have decreased. This indicates that for difficult samples, the Mamba model tends to associate certain incomprehensible regions in the external states.

**Flaws in Internal State Correlation.**  To unveil patterns in internal state correlations and detect flaws within them, we also conducted a comparative analysis of the internal correlation scores within the Mamba model, as depicted in Figure 4(c) and (d). For both simple and difficult samples, the internal correlation score of state $x_n^{(\ell)}$ is generally better than that of other states. Furthermore, simultaneous comparison of simple and difficult samples reveals a decrease in the internal correlation scores of all states, including state $x_n^{(\ell)}$. This suggests that for difficult samples, the Mamba model tends to correlate with some incomprehensible regions within the internal states.

## 4   How to Repair Flaws?

In principle, repairing flaws within the model's internals can enhance Mamba's decision-making process and improve its performance. However, there has been limited research specifically addressing such issues in Mamba models, particularly when applied in the domain of visual processing. Therefore, in this section, we investigate post-hoc flaw repair in the Mamba model from two perspectives: external state correlation and internal state correlation.

### 4.1   External State Correlation Repair

To repair flaws related to external state correlations, it is essential to identify the key components within the Mamba block that need fixing. In flaw identification regarding external state correlations, it is observed that the states $x_n^{(\ell)}$, $c_n^{(\ell)}$, $s_n^{(\ell)}$, and $z_n^{(\ell)}$ exhibit flaws when predicting difficult samples. For these states, the external correlation scores for simple samples are higher than those for difficult samples. This implies that the model primarily correlates with foreground regions in correct predictions and with background regions in incorrect predictions. Furthermore, considering that external state interactions occur only within Conv and SSM modules (as indicated by Eqn.(2) and Eqn.(3)), we empirically suggest that the Conv and SSM components are crucial for influencing the model's anomalous decisions. Theoretically, it is also feasible to apply external flaw repair to other states.

We focus on repairing the external correlation flaws of the states $c_n^{(\ell)}$ and $s_n^{(\ell)}$ output by the Conv and SSM modules in the deeper blocks. Specifically, we first identify difficult samples from the training set and then constrain the external correlations of the hidden states $c_n^{(\ell)}$ and $s_n^{(\ell)}$ using foreground annotations $m$:

$$\text{Loss}_\mathbf{e} = \mathbb{E}_{HW}(\mathbf{e}^{(\ell,c)} \odot m) + \mathbb{E}_{HW}(\mathbf{e}^{(\ell,s)} \odot m), \tag{11}$$

where $\mathbb{E}_{HW}$ denotes the averaging operation over the two-dimensional matrix. It is important to note that during backpropagation, each term in the computational graph of $\text{Loss}_\mathbf{e}$ is differentiable, which involves second-order gradients as specified in Eqn. (7). The complete loss function, combining with the original task loss, is formulated as follows:

$$\text{Loss} = \text{Loss}_\mathbf{ce} + \lambda \text{Loss}_\mathbf{e}, \tag{12}$$

where $\text{Loss}_\mathbf{ce}$ denotes the cross-entropy loss for the image recognition task, and $\lambda$ serves to balance the magnitudes of the respective loss components.

## 4.2 Internal State Correlation Repair

Similarly, when identifying flaws related to internal state correlations, it is observed that states in different blocks exhibit flaws when predicting difficult samples. For instance, the internal correlation scores of state $x_n^{(\ell)}$ for simple samples are higher than those for difficult samples. This suggests that in predictions biased toward correctness within a class, the model aligns with internally consistent regions of the state, characterized by lower overall complexity. Conversely, in predictions biased toward incorrectness within a class, the model tends to focus on internally inconsistent regions of the state with higher overall complexity. In our experiments, we consider the linear mapping $W_s^{(\ell)}$ within deeper blocks as a critical component influencing the model's predictions. However, theoretically, it is also feasible to apply internal flaw repair to other states.

We focus on repairing the internal correlation flaws of the states $x_n^{(\ell)}$ output by the linear mapping $W_x^{(\ell)}$ within deeper blocks. Specifically, we first select $J$ simple samples for each class from the training set and create corresponding internal correlation templates $\hat{\mathbf{i}}_n^{(\ell,x)+} = 1 - (\frac{1}{J} \sum_{j=1}^{J} \mathbf{i}_{n,j}^{(\ell,x)+})$ for each class. We then utilize these templates to constrain the internal correlations of $x_n^{(\ell)}$:

$$\text{Loss}_\mathbf{i} = \mathbb{E}_D(\mathbf{i}_n^{(\ell,x)} \odot \hat{\mathbf{i}}_n^{(\ell,s)+}). \tag{13}$$

Similarly, during backpropagation, each term in the computational graph of $\text{Loss}_\mathbf{i}$ is also differentiable, which involves second-order gradients as specified in Eqn. (9). Combined with the loss for the original task, the complete loss is as follows:

$$\text{Loss} = \text{Loss}_\mathbf{ce} + \gamma \text{Loss}_\mathbf{i}, \tag{14}$$

where $\gamma$ is a balancing factor to adjust the scale between loss functions.

It is important to note that external and internal state corrections represent two distinct perspectives, allowing these two methods to be used orthogonally. To maximize flaw repair, we recommend sequentially repairing external flaws (Eqn. (11)) followed by internal flaws (Eqn. (13)).

## 4.3 Results of Flaw Repair

We evaluated the proposed flaw repair method on five state-of-the-art Mamba models for visual tasks, including ViM-T [17], VMamba-T [18], SiMBA-S [21], EfficientVMamba-T (EMamba-T) [34], and LocalVim-T [35]. To ensure smooth operation of the Mamba models within limited computational resources, we modified certain model parameters, such as the number of blocks and the dimensionality of hidden states. Additionally, we conducted experiments on three scales of the ImageNet [36] dataset: ImageNet-50, ImageNet-300, and ImageNet-1K. To obtain the image foreground annotations $m$ as defined in **Definition 1** from the ImageNet dataset, we utilized annotations from the ImageNet-S dataset [37]. These annotations correspond to those used during training, covering all 50 classes in ImageNet-50, 300 classes in ImageNet-300, and 919 classes in ImageNet-1K, with an average of 10 foreground-labeled samples per class. Further details on experimental settings are provided in Appendix B.

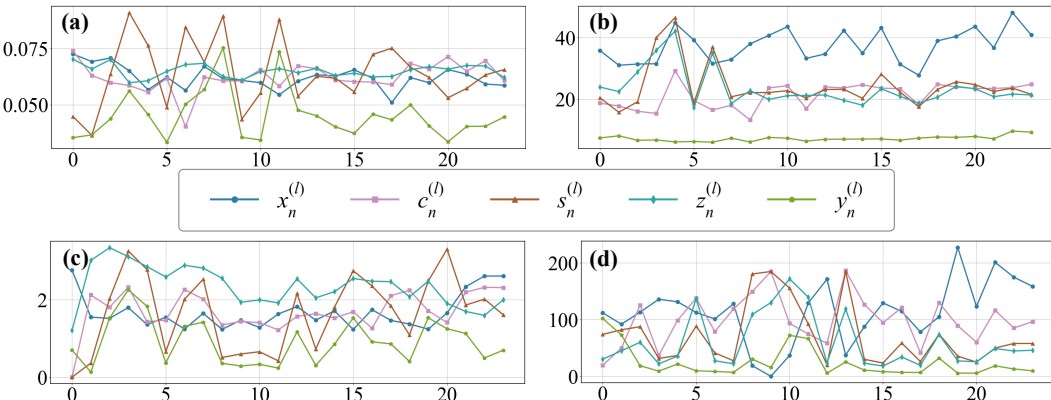

Figure 5: Comparison of external and internal state correlation scores across different blocks in the Mamba model before and after flaw repair. **(a)** and **(b)** show the external and internal state correlation scores before flaw repair, respectively. **(c)** and **(d)** present the external and internal state correlation scores after flaw repair, respectively.

Table 1: Comparison of model accuracy after external state flaw repair for different states. Taking '$x_n^{(\ell)}$' as an example, it represents the external flaw repair of state $x_n^{(\ell)}$. The results in the second row of the table correspond to external flaw repair, while the third row presents the outcomes of internal flaw repair. The experiment was conducted within the last block of the ViM model, using the ImageNet-50 dataset.

| **Base** | $x_n^{(\ell)}$ | $c_n^{(\ell)}$ | $s_n^{(\ell)}$ | $z_n^{(\ell)}$ | $y_n^{(\ell)}$ | $x_n^{(\ell)}+c_n^{(\ell)}$ | $x_n^{(\ell)}+s_n^{(\ell)}$ | $x_n^{(\ell)}+z_n^{(\ell)}$ | $c_n^{(\ell)}+s_n^{(\ell)}$ |
|---|---|---|---|---|---|---|---|---|---|
| 76.44 | 78.40 | 78.44 | 78.28 | 77.96 | 78.28 | 78.56 | 78.04 | 78.04 | 78.48 |
| 76.44 | 78.20 | 79.16 | 78.88 | 78.32 | 78.32 | 77.72 | 78.76 | 78.44 | 78.42 |

**Comparing State Correlation Scores Before and After Repair.** To validate the effectiveness of the flaw repair methods, we compared the internal and external state correlation scores of difficult samples before and after flaw repair, as shown in Figure 5. It can be observed that the proposed flaw repair methods effectively enhance the correlation scores of the Mamba model, thereby improving the predicted correlation regions within both internal and external states. Specifically, for the ViM model, after external state flaw repair, the external correlation scores of the states $c_n^{(\ell)}$ are significantly higher than before repair. Similarly, after internal state flaw repair, the internal correlation scores of the states $x_n^{(\ell)}$ are also notably higher than before repair.

**Repairing External and Internal Flaws in Different States.** As shown in Table 1 (second row), we conducted experiments on external flaw repair for different states within the same block. The results indicate that simultaneously performing external flaw repair on states $c_n^{(\ell)}$ and $s_n^{(\ell)}$ leads to an improvement in model accuracy. This aligns with our findings in the main text regarding flaw detection, where states $x_n^{(\ell)}$, $c_n^{(\ell)}$, $s_n^{(\ell)}$, and $z_n^{(\ell)}$ exhibit flaws when predicting challenging samples. Furthermore, since the Conv and SSM modules in each block facilitate external interactions, applying external flaw repair to states $c_n^{(\ell)}$ and $s_n^{(\ell)}$ is effective.

Similarly, as illustrated in Table 1 (third row), we performed experiments on internal flaw repair for different states within the same block. The results demonstrate that individually addressing internal flaws in state $x_n^{(\ell)}$ also results in improved model accuracy. This supports our earlier findings regarding flaw detection, where state $x_n^{(\ell)}$ shows flaws when predicting challenging samples, thus validating the effectiveness of internal flaw repair for state $x_n^{(\ell)}$.

**Repairing SOTA Vision Mamba Models.** As shown in the Table 2, we validated the proposed flaw repair methods on various mainstream Vision Mamba models. It can be observed that regardless of whether external or internal state flaw repair is performed, the accuracy of the repaired models exceeds

Table 2: Comparison of the accuracy of the SOTA Vision Mamba model after flaw repair. Base' denotes the original model, +Ext', +Int', and +All' represent models after external flaw repair, internal flaw repair, and simultaneous external and internal flaw repair, respectively.[2]

|        | ViM-T      | VMamba-T   | SiMBA-S    | EMamba-T   | LocalVim-T |
|--------|------------|------------|------------|------------|------------|
| **ImageNet-50** | | | | | |
| **Base** | 76.44      | 79.96      | 81.32      | 81.44      | 75.08      |
| **+Ext** | 78.48+2.04 | 81.08+1.12 | 84.48+3.16 | 82.68+1.24 | 78.68+3.60 |
| **+Int** | 78.20+1.76 | 82.36+2.40 | 84.64+3.32 | 83.24+1.80 | 78.20+3.12 |
| **+All** | 79.68+3.24 | 82.28+2.32 | 86.52+5.20 | 83.32+1.88 | 80.56+5.48 |
| **ImageNet-300** | | | | | |
| **Base** | 75.11      | 75.04      | 68.08      | 74.67      | 70.39      |
| **+Ext** | 77.87+2.76 | 76.01+0.97 | 69.73+1.65 | 75.03+0.36 | 73.09+2.70 |
| **+Int** | 77.76+2.65 | 76.31+1.27 | 69.93+1.85 | 75.55+0.88 | 72.71+2.32 |
| **+All** | 79.53+4.42 | 76.75+1.71 | 70.58+2.50 | 75.66+0.99 | 74.84+4.45 |
| **ImageNet-1K** | | | | | |
| **Base** | 71.64      | 67.54      | 51.06      | 67.35      | 57.70      |
| **+Ext** | 73.02+1.38 | 68.34+0.80 | 52.12+1.06 | 67.62+0.27 | 59.30+1.60 |
| **+Int** | 72.79+1.15 | 68.67+1.13 | 52.23+1.17 | 67.89+0.54 | 59.90+2.20 |
| **+All** | 73.30+1.66 | 68.68+1.14 | 52.24+1.18 | 67.84+0.49 | 60.83+3.13 |

that of the original models. Specifically, for external state flaw repair, on ImageNet-50, the accuracy of VMamba increased by 1.12% after flaw repair, and the accuracy of SiMBA-S increased by 3.16%. For internal state flaw repair, on ImageNet-1K, the accuracy of ViM increased by 1.15% after flaw repair, and the accuracy of LocalViM increased by 2.20%. It is worth noting that external state flaw repair and internal state flaw repair can be orthogonal. For example, on ImageNet-50, simultaneously performing external state flaw repair and internal state flaw repair on ViM increased the accuracy by 2.04% and 1.76%, respectively, compared to performing each repair method individually. This resulted in an overall improvement of 3.24% compared to the original model.

## 5   Related Works

**Model Optimization.**   Optimizing model performance through various prior designs or theoretical derivations has always been a pursuit in the field of artificial intelligence. Existing methods for optimizing the Mamba model, especially when applied to visual tasks, mainly involve adjusting the network architecture [17–22]. For instance, improvements have been made through the bidirectional scanning mechanism [17], cross-scanning mechanism [18], continuous 2D scanning mechanism [19], and incorporating new channel modeling techniques [21]. However, these enhancements are pre-designed and require extensive trial and error. For a more detailed understanding of Mamba's applications in computer vision, we recommend readers refer to recent surveys by Zhang et al. [14] and Xu et al. [16]. So far, there has been no research focusing on post-hoc optimization to correct internal flaws within the Mamba model to improve its performance. Optimizing models beyond Mamba, such as Convolutional Neural Networks [1, 38] and Transformers [3, 39], is also primarily achieved through architectural improvements [2, 40–42, 4], feature enhancement [43–46], or some post-hoc debugging methods [47–50]. However, these approaches either cannot be directly applied to a brand-new Mamba model or do not involve post-hoc optimization. In summary, unlike the aforementioned studies, the proposed Vision Mamba Mender is the first framework dedicated to post-hoc analysis and optimization of the Mamba model. Its aim is to rectify flaws within Mamba models in the domain of vision, thereby further enhancing the performance of Mamba models.

---

[2]We adjusted the experimental setup and refined methodological details to enhance the fairness and reproducibility of the comparisons. Detailed experimental settings can be found in Appendix B.

**Model Explanation.** Enhancing model transparency and trustworthiness through research on model explainability has been a major focus in the field of artificial intelligence. In the context of Mamba model explainability, most studies have primarily concentrated on the model's context learning capabilities [26], factual recall abilities [27], and comparisons of the explainability between Mamba and previous models [28]. These studies do not address the identification of flaws in the Mamba model's mechanisms and are focused on natural language processing (NLP), leaving a gap in their application to visual tasks. In the realm of computer vision, Ali et al.[29] established a connection between the selective SSM in the Mamba block and the self-attention mechanism in Transformers. They developed a feature attribution method for Mamba based on Attention-Rollout[51] and Transformer-Attribution [31], which can be used for image feature attribution. However, this method is limited to the selective SSM module and does not consider or apply to other modules within the Mamba model. In research on model explainability beyond the Mamba model, a plethora of methods have emerged, including activation-based methods [52–56], gradient-based methods [30, 57, 33], LRP-based methods [58, 59, 31], and perturbation-based methods [60–62]. However, most of these explainability methods are designed for specific network architectures and may not be directly applicable to Mamba. Inspired by Grad-CAM [30], this paper introduces Grad-ESC and Grad-ISC methods tailored for the Mamba's external and internal states, respectively. These methods effectively establish the correlation between the model's states and predictions, providing simple yet powerful analytical tools for identifying flaws in the Mamba model's mechanisms.

## 6 Discussion

As one of the most prominent models today, Mamba has found widespread application in the field of computer vision. While attitudes towards its utilization in visual tasks may vary, it does not impede the exploration of its novel internal operational mechanisms. On the contrary, it urges further research into optimizing the performance of Mamba models applied in visual tasks. Moreover, at this juncture, considering the potential changes in future architectures, devising a set of architecture-agnostic post-optimization methods becomes crucial.

However, in this work, we only analyzed the overall operational mechanisms of each module within Mamba, without delving into the internal details of each module. For instance, we hypothesize that there might be flaws within the SSM module itself during prediction, and further optimization of its internal details may potentially enhance the model's performance. Additionally, similar post-optimization paradigms should be further explored in non-visual tasks within Mamba or even applied in non-Mamba architectures.

## 7 Conclusion

Mamba incorporates intricate computational modules, making it non-trivial to delve into its operational mechanisms. In this paper, we have proposed a post-analysis and post-optimization approach to understand the workings of Mamba in visual recognition tasks and enhance its performance. Specifically, departing from existing pre-defined optimization methods, we have introduced predictive correlation analysis and correlation scoring definitions from both internal and external perspectives of Mamba's states. These methodologies aim to identify flaws within Mamba's operational mechanisms. Simultaneously, we have presented corresponding approaches for repairing internal and external flaws to optimize model performance. Extensive and comprehensive experiments have demonstrated the effectiveness of Vision Mamba Mender on mainstream Mamba architectures.

## Acknowledgments and Disclosure of Funding

This research was supported by the Joint Funds of the Zhejiang Provincial Natural Science Foundation of China under Grant No. LHZSD24F020001 and Ningbo Natural Science Foundation (2022J182).

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

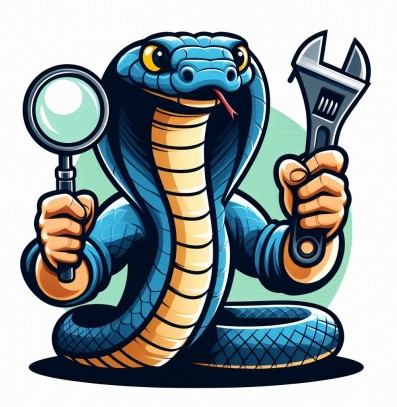

A Vision Mamba Mender. The magnifying glass held in one hand symbolizes flaw detection, while the wrench held in the other hand symbolizes flaw repair.

# Appendix for Vision Mamba Mender

To facilitate a better understanding of the value and significance of this work, as well as to thoroughly demonstrate the effectiveness and applicability of the proposed method, we have provided the algorithm code in supplementary material. This code will be made publicly available. Additionally, in the appendix, we have supplemented more information about the Mamba architecture, related work on Mamba interpretability, detailed experimental settings, additional results on state flaw identification, additional results on state flaw repair, and detailed ablation experiments, as follows.

## A  Origin of the Mamba Model

One of the core components of Mamba is the State Space Model (SSM). SSMs were initially inspired by continuous systems, which map a one-dimensional input signal $x(t) \in \mathbb{R}$ to a one-dimensional output signal $y(t) \in \mathbb{R}$ through an $N$-dimensional latent state $h(t) \in \mathbb{R}^N$. The general computational process can be defined as follows:

$$
\begin{aligned}
h'(t) &= \mathbf{A}h(t) + \mathbf{B}x(t), \\
y(t) &= \mathbf{C}h(t),
\end{aligned}
\tag{15}
$$

where $\mathbf{A} \in \mathbb{R}^{N \times N}$, $\mathbf{B} \in \mathbb{R}^{N \times 1}$, and $\mathbf{C} \in \mathbb{R}^{1 \times N}$ are the state matrix, input matrix, and output matrix, respectively.

Unlike the continuity in Equation 15, Structured State Space Models (S4) [11] employ transformation methods, such as the zero-order hold (ZOH), to discretize it.

$$
\begin{aligned}
h'(t) &= \overline{\mathbf{A}}h(t) + \overline{\mathbf{B}}x(t), \\
y(t) &= \overline{\mathbf{C}}h(t),
\end{aligned}
\tag{16}
$$

where $\overline{\mathbf{A}} = \exp\left(\Delta \mathbf{A}\right)$ and $\overline{\mathbf{B}} = (\Delta \mathbf{A})^{-1}(\exp\left(\Delta \mathbf{A}\right) - \mathbf{I}) \cdot \Delta \mathbf{B}$ are the discretized parameters, and $\Delta$ denotes the step size.

Building on this foundation, the Selective State Space Model (S6) creatively introduces selective scanning to overcome the limitations imposed by time-invariant parameters on context representation learning:

$$
\mathbf{B} = \mathcal{S}_B(\mathbf{x}), \, \mathbf{C} = \mathcal{S}_C(\mathbf{x}), \, \Delta = \tau_\Delta(\Delta + S_\Delta(\mathbf{x})),
\tag{17}
$$

where $S_B$, $S_C$, and $S_\Delta$ are different linear mappings. Additionally, for more detailed information on Mamba's specifics and applications, we recommend readers consult several recent and well-regarded review articles [63, 64, 14–16].

# B  Detailed Experimental Settings

To facilitate the reproducibility of our results, we provide detailed experimental settings as follows. Throughout the entire experiment, we utilized 8 NVIDIA A40 GPU cards and a CPU with 24 cores and 500GB of memory.

**Model Parameter Settings.**  To enable the Mamba model to train efficiently within limited computational resources, we adjusted certain parameters across the Mamba models. For instance, in the case of VMamba-T, we set the patch size to 16x16. For SiMBA-S, the model depth was adjusted to [2, 3, 3, 2], and we introduced a class token for classification in the last block. For EfficientVMamba-T, we similarly set the patch size to 16x16. In the case of LocalViM-T, we reduced the model depth from 20 to 9 and set the state dimensionality to 128. These parameter adjustments were made to validate the proposed methods under constrained resources. While this may somewhat reduce the baseline performance, it does not affect the fairness of comparing the Mamba model's performance before and after flaw identification and repair.

**Model Training Settings.**  To ensure a fair comparison with limited resources, our training settings primarily followed the experimental setup of DeiT [43]. Specifically, we employed data augmentation techniques such as random cropping and random horizontal flipping. When training on 224x224 input images, we optimized the model using AdamW [65] with a momentum of 0.9, a total batch size of 128, and a weight decay of 0.1. We utilized a cosine learning rate schedule with an initial learning rate of 5e-4, training the Mamba model for 300 epochs. In particular during the baseline model training, these training strategies resulted in an exceedingly smooth training curve in the later stages, effectively optimizing the fit. During testing, we performed center cropping on the validation set to extract 224x224 images.

**State Flaw Identification.**  For external state correlation analysis, the threshold $\alpha$ is set to 0.5 by default. For internal state correlation, the threshold $\beta$ is set to 0.3 by default. The impact of thresholds on flaw detection can be found in the Appendices C and D. It is worth noting that for multi-branch Mamba, when conducting external flaw identification and repair, we merge all branches corresponding to the same state together. However, for internal flaw identification and repair, we consider each branch of the state separately.

**State Flaw Repair.**  In our experiments, the balance weight $\lambda$ for the loss function of external state flaw repair is set to $1e + 7$ by default, and the balance weight $\gamma$ for the loss function of internal state flaw repair is also set to $1e + 7$ by default. Furthermore, based on the conclusions about flaw identification in the main text, external state flaw repair is applied by default to the first Mamba block, while internal state flaw repair is applied to the last Mamba block. The impact of repairing flaws in different blocks on model performance can be found in Appendices E and F.

# C  Impact of Threshold $\alpha$ on External State Correlation Scores

As shown in Figure 6, we compared the impact of different values of threshold $\alpha$ on external state correlation scores. By contrasting simple samples with difficult samples, we found that the value of $\alpha$ does not significantly affect the relative magnitudes of the external correlation scores. Specifically, regardless of whether $\alpha$ is set to 0.0, 0.2, or 0.8, the external correlation scores for states $x_n^{(\ell)}$, $c_n^{(\ell)}$, $s_n^{(\ell)}$, $z_n^{(\ell)}$, and $y_n^{(\ell)}$ in difficult samples all decrease compared to those in simple samples. Additionally, when $\alpha$ is set between 0.4 and 0.6, more details can be observed; in different blocks, the external correlation scores of states $x_n^{(\ell)}$, $c_n^{(\ell)}$, $s_n^{(\ell)}$, and $z_n^{(\ell)}$ are better than those of state $y_n^{(\ell)}$. Therefore, we recommend that the value of $\alpha$ should not be too extreme when calculating external state correlation scores. It can be set between 0.4 and 0.6; we use a default setting of $\alpha = 0.5$.

# D  Impact of Threshold $\beta$ on Internal State Correlation Scores

We also examined the impact of different values of threshold $\beta$ on internal state correlation scores, as shown in Fig. 7. Similarly, by comparing simple and difficult samples, it can be observed that the value of $\beta$ does not affect the relative magnitudes of the internal correlation scores. Specifically,

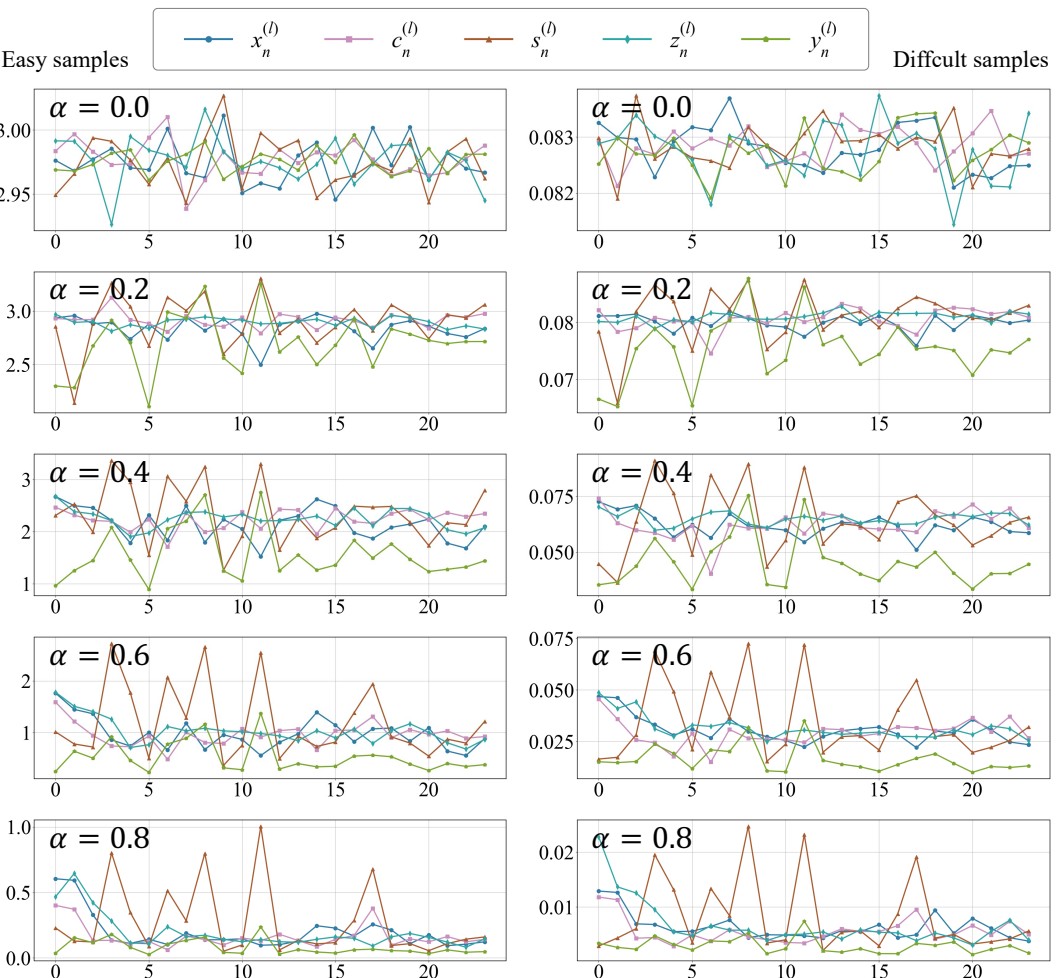

Figure 6: Comparison of external state correlation scores for simple and difficult samples under different threshold values of $\alpha$. The left column of images shows the external state correlation scores for simple samples at various $\alpha$ values. The right column of images shows the external state correlation scores for difficult samples at various $\alpha$ values.

regardless of whether $\beta$ is set to 0.0, 0.2, or 0.8, the internal correlation scores for states $x_n^{(\ell)}$, $c_n^{(\ell)}$, $s_n^{(\ell)}$, $z_n^{(\ell)}$, and $y_n^{(\ell)}$ in difficult samples all decrease compared to those in simple samples. Additionally, when $\beta$ is set between 0.4 and 0.6, more details can be observed; in different blocks, the internal correlation scores of state $x_n^{(\ell)}$ are better than those of states $c_n^{(\ell)}$, $s_n^{(\ell)}$, $z_n^{(\ell)}$, and $y_n^{(\ell)}$. Therefore, we recommend that the value of $\beta$ should not be too extreme when calculating internal state correlation scores. It can be set between 0.4 and 0.6; we use a default setting of $\beta = 0.5$.

## E External Flaw Repair in Different Blocks

As shown in Figure E(a), we conducted a study on repairing external flaws in different blocks, revealing a clear enhancement in model performance following external flaw repair. Specifically, when external flaws are repaired in the final block, the model's accuracy improves by 2.04% compared to the baseline. In contrast, while repairing external flaws in other blocks, such as the tenth block, also yields an accuracy increase, the improvement is only 1.72%. This discrepancy may be attributed to the presence of external flaws in the states across nearly every block, as illustrated in Figure 2, including states such as $x_n^{(\ell)}$, $c_n^{(\ell)}$, $s_n^{(\ell)}$, and $z_n^{(\ell)}$, which exhibit significant external flaws on difficult samples. However, due to structures like residual connections, the influence of external flaws on model predictions varies across different blocks.

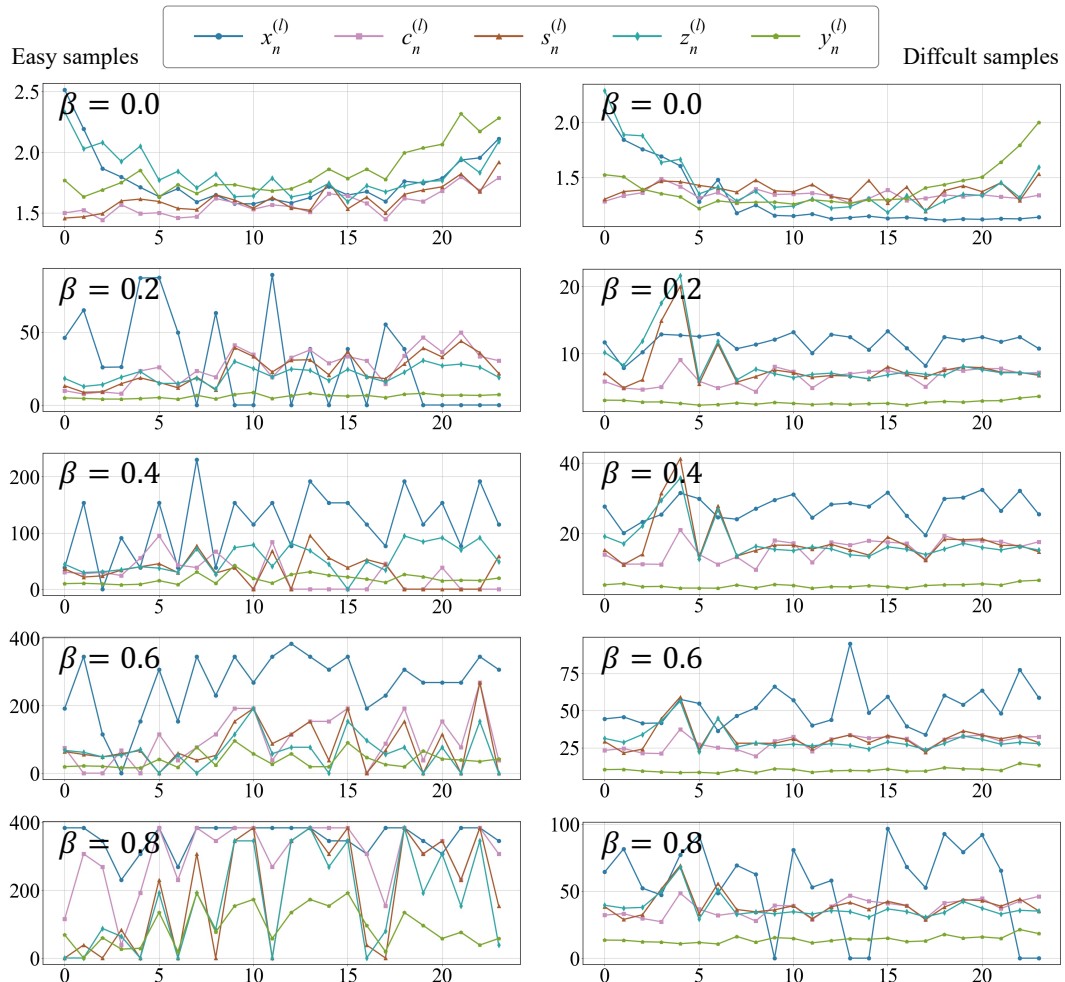

Figure 7: Comparison of internal state correlation scores for simple and difficult samples under different threshold values of $\beta$. The left column of images shows the internal state correlation scores for simple samples at various $\beta$ values. The right column of images shows the internal state correlation scores for difficult samples at various $\beta$ values.

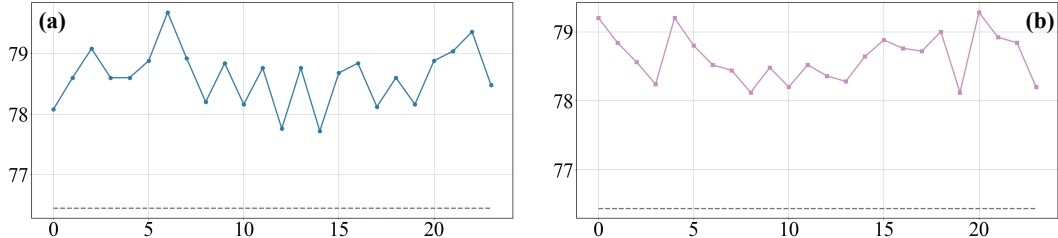

Figure 8: Comparison of model accuracy after state flaw repair in different blocks. (a) shows the results of external state flaw repair, conducted on states $c_n^{(\ell)}$ and $s_n^{(\ell)}$ in each block of the ViM model, using the ImageNet-50 dataset. (b) shows the results of internal state flaw repair, conducted on state $x_n^{(\ell)}$ in each block of the ViM model, also using the ImageNet-50 dataset.

## F   Internal Flaw Repair in Different Blocks

As shown in Figure E(b), we conducted a study on repairing internal flaws in different blocks, which also demonstrates a noticeable improvement in model performance. Specifically, when internal flaws are repaired in the final block, the model's accuracy increases by 1.76% compared to the baseline. In contrast, repairing internal flaws in other blocks, such as the second-to-last block, results in an accuracy improvement of 2.4%. Furthermore, repairing internal correlations in other blocks yields varying levels of accuracy enhancement. This variation may similarly stem from the model's internal structures, such as residual connections, which lead to different degrees of influence from internal flaws on model performance across the various blocks.

