# OpenReview forum: "Vision Mamba Mender"
_NeurIPS.cc/2024/Conference — NeurIPS 2024 poster_

### Official Review · Reviewer_JDBm · 2024-07-04

**Soundness:** 3
**Presentation:** 3
**Contribution:** 3
**Rating:** 6
**Confidence:** 4

**Summary:**

The paper introduces a novel post-hoc optimization strategy for existing Vision Mamba architectures, termed Vision Mamba Mender, aimed at enhancing the performance of Mamba models in visual recognition tasks. The authors seek to identify and rectify flaws in the Mamba model’s mechanisms from both external and internal hidden state perspectives, proposing a state correlation analysis method. Through this analysis, the authors pinpoint critical modules requiring repair and subsequently design a state correlation constraint method. The authors conduct extensive experiments on several Vision Mamba architectures to validate the efficacy of the proposed optimization strategy.

**Strengths:**

1. The paper proposes a novel and intriguing method to enhance state space models from a post-hoc perspective for Mamba. The state correlation analysis used to identify flaws in Mamba, and the subsequent state correlation constraint employed to rectify these flaws, are both new explorations for this kind of technique.
2. The proposed method for identifying and rectifying flaws in Mamba is systematic to some degree. Moreover, the experiments are clearly designed and well-executed, demonstrating that the proposed method can be applied to mainstream Mamba architectures, further improving their performance.
3. The paper is well-written, with clear language and a well-structured presentation. The background and motivation are clearly articulated and convincing. Each part of the method is presented step-by-step and in detail, allowing readers to effectively grasp the core ideas.
4. The paper also provides some insights into the working mechanisms of Mamba, having the potential to inspire future research on the explanation of Mamba models.

**Weaknesses:**

1. Before reading the methodology section, the description of "certain modules of Mamba" (Lines 65-67) may be confusing to readers. The authors should avoid using such vague terms.
2. The results in Table 1 show that the proposed method improves the performance of Mamba by 1% to 5%. For the sake of rigor, the description in Line 71, stating that the performance is "enhanced by 3% to 5%," does not seem very precise.
3. In Lines 224-229, the authors claim that the states $c_n^{(l)}$ and $s_n^{(l)}$ in the shallowest blocks are crucial for influencing the model’s anomalous decisions. However, this cannot be observed from Fig. 4, as $c_n^{(l)}$ and $s_n^{(l)}$ do not appear to vary most significantly from the simple sample (Fig. 4a) to the hard sample (Fig. 4b).
4. In addition to comparing the state correlation scores before and after flaw repair (Fig. 5), the authors should provide more visual comparisons of the state correlation (like Fig. 2 and Fig. 3), which would be more intuitive for the readers.
5. Only analyzing and enhancing state space models from a post-hoc perspective for Mamba may not be enough.

**Questions:**

1. In Eqns. (8) and (10), the two terms are multiplied together rather than averaged. What is the rationale behind this design choice?
2. In Figure 4, why are both the ViM and VMamba structures adopted to obtain these observations?
3. The loss function defined in Eq. (11) seems incorrect. Could the authors clarify this?
4. In Table 1, it appears that the performance of "Internal Flaw Repair" is better than that of "External Flaw Repair." Are the authors aware of this, and do they have any thoughts on why this is the case?

**Limitations:**

The limitations appear to have been discussed in Section 6 by the authors. Moreover, there is no societal impact on the work performed.

---

> ### Author Rebuttal · Authors · 2024-08-07
>
> We appreciate the reviewer’s recognition of the novelty and interest of our proposed method and the acknowledgment of its contribution to the interpretability research of the Mamba model. Below are our responses to each of your comments.
>
> ---
>
> > **Q1**: Before reading the methodology section, the description of "certain modules of Mamba" (Lines 65-67) may be confusing to readers. The authors should avoid using such vague terms.
> >
>
> **A1**: We apologize for the confusion caused by this description. Lines 65-67 refer to "certain modules of Mamba," which specifically denotes modules within a Mamba block that exhibit defects after state correlation analysis. For example, the module identified with defects through internal state correlation analysis is $x_{n}^{(l)}$. We will update this section in the paper to clarify this description and avoid any confusion.
>
> ---
>
> > **Q2**: The results in Table 1 show that the proposed method improves the performance of Mamba by 1% to 5%. For the sake of rigor, the description in Line 71, stating that the performance is "enhanced by 3% to 5%," does not seem very precise.
> >
>
> **A2**: We apologize for this typographical error and have corrected the description in Line 71.
>
> ---
>
> > **Q3**: In Lines 224-229, the authors claim that the states  $c_n^{(l)}$ and $s_n^{(l)}$  in the shallowest blocks are crucial for influencing the model’s anomalous decisions. However, this cannot be observed from Fig. 4, as $c_n^{(l)}$ and $s_n^{(l)}$  do not appear to vary most significantly from the simple sample (Fig. 4a) to the hard sample (Fig. 4b).
> >
>
> **A3**: Yes, you are correct. Figure 4a and 4b only show that for hard samples, the external state correlation scores decrease across all states. However, since only Conv and SSM participate in external interactions with states within a Mamba block (lines 121-123 of the paper), we believe that states $c_n^{(l)}$ and $s_n^{(l)}$ have the most significant impact on the model’s defective decisions. It is important to note that subsequent ablation experiments show that defect repair on states $c_n^{(l)}$ and $s_n^{(l)}$ leads to improved model performance (Appendix E).
>
> ---
>
> > **Q4**: In addition to comparing the state correlation scores before and after flaw repair (Fig. 5), the authors should provide more visual comparisons of the state correlation (like Fig. 2 and Fig. 3), which would be more intuitive for the readers.
> >
>
> **A4**: Thank you for your suggestion. We found that, in addition to the improvement in state correlation scores after flaw repair (as shown in Figure 5), the visual morphology of the states, similar to Figures 2 and 3, also changes. Specifically, after defect repair, the difficult samples in Figure 2 exhibit a greater focus on the foreground compared to before repair. We will include these additional visual comparisons in the appendix of the paper.
>
> ---
>
> > **Q5**: Only analyzing and enhancing state space models from a post-hoc perspective for Mamba may not be enough.
> >
>
> **A5**: Thank you for your insightful comment; we understand your concerns. In addition to post-hoc analysis and enhancement of the Mamba model, there are indeed preemptive approaches to strengthen the Mamba model's architecture, such as improving its scanning mechanisms. However, these methods often require strong priors and extensive trial-and-error (lines 32-38 of the paper).
>
> The primary distinction of our work, compared to existing studies, lies in its post-hoc analysis and optimization of the Mamba model. This approach not only provides a clear understanding of Mamba's operational mechanisms but also allows for targeted and precise solutions to the issues encountered during inference. We believe that the insights gained from post-hoc analysis will better assist researchers in designing more refined Mamba architectures, which is one of the key contributions of this work.
>
> Moreover, our paper goes beyond analyzing and enhancing the state space model operations within Mamba blocks by also addressing various linear mapping operations, including general 1D convolutions and gating mechanisms within the Mamba blocks. This analysis and enhancement are comprehensive and thorough.
>
> We hope this response addresses your concerns. Exploring additional optimization methods for the Mamba model and enhancing the applicability of our proposed methods to more complex tasks will be a continued focus of our future research.

---

> ### Author Response · Authors · 2024-08-07
> **Rebuttal by Authors [Q6-Q9]**
>
> > **Q6**: In Eqns. (8) and (10), the two terms are multiplied together rather than averaged. What is the rationale behind this design choice?
> >
>
> **A6**: The rationale behind this design choice is that the units and scales of the terms in these equations are different. Averaging them could lead to information loss and might not adequately reflect the importance of each term. Multiplying the terms amplifies the differences between them and highlights the significance of each term more effectively.
>
> ---
>
> > **Q7**: In Figure 4, why are both the ViM and VMamba structures adopted to obtain these observations?
> >
>
> **A7**: We sincerely apologize for the confusion regarding the description of Figure 4. In reality, all observations in Figure 4 are based on the ViM structure. As detailed in lines 196-198 and 204-206 of the paper, Figure 4(a,b) shows the external state correlation scores for different states within Mamba blocks for simple and hard samples, respectively. Similarly, Figure 4(c,d) displays the internal state correlation scores for these states in the same Mamba blocks for simple and hard samples. The x-axis of each subplot represents different Mamba blocks, and the y-axis indicates the magnitude of the correlation scores. We will correct the description of Figure 4 in the paper.
>
> ---
>
> > **Q8**: The loss function defined in Eq. (11) seems incorrect. Could the authors clarify this?
> >
>
> **A8**: We apologize for the typographical error. The correct loss function should be: $\text{Loss}\_{\mathbf{e}} = \mathbb{E}\_{HW}(\mathbf{e}^{(\ell,c)+} \odot m) + \mathbb{E}\_{HW}(\mathbf{e}^{(\ell,s)+} \odot m)$ We have corrected this in the paper. Thank you for pointing this out.
>
> ---
>
> > **Q9**: In Table 1, it appears that the performance of "Internal Flaw Repair" is better than that of "External Flaw Repair." Are the authors aware of this, and do they have any thoughts on why this is the case?
> >
>
> **A9**: Great question. The reason why "Internal Flaw Repair" shows better performance than "External Flaw Repair" is primarily due to the more pronounced patterns observed in internal state correlations compared to external state correlations. Specifically, as shown in Figure 4(c,d), the internal state correlation scores for states such as $x_{n}^{(\ell)}$ drop more significantly for difficult samples. Therefore, repairing internal state correlations for $x_{n}^{(\ell)}$ results in more substantial improvements.

---

> > ### Comment · Reviewer_JDBm · 2024-08-11
> > **Thanks for response**
> >
> > Thank you for your response. My concerns have been mainly addressed. I have also reviewed the comments from other reviewers and the corresponding responses. I have increased my score to 6.

---

> ### Author Response · Authors · 2024-08-12
>
> Thank you for your positive feedback. We are pleased that your concerns have been addressed. We greatly appreciate your support for our work.

---

### Official Review · Reviewer_3tmX · 2024-07-10

**Soundness:** 3
**Presentation:** 3
**Contribution:** 3
**Rating:** 4
**Confidence:** 4

**Summary:**

The papers addresses the limitations of Mamba based models in vision through a post-hoc optimization scheme that address external state flaws and internal state flaws, identified through respective internal and external state correlation analysis. The introduced corrective measures improve performance on image classification tasks across various mamba-based backbones on the imagenet dataset.

**Strengths:**

1. The paper is very well-motivated and addresses a specific topic very relevant to the vision community.
2. The presented correlation analysis is novel and detailed, and the resulting flaws identified along with presented mitigations are consistent and show consistent improvements across various backbones.
3. The paper is well-written and easy to follow.

**Weaknesses:**

There are two major concerns here:

1. The method is limited by the required annotations (for example foreground annotation) which limits its usefulness across other classification datasets and other tasks.
2. The paper does not evaluate on any detection/segmentation benchmarks. This is required to understand whether the features learned by the backbone are rich enough to generalize to these more complex tasks. All recent vision backbones, including the various mamba-based ones, report detection results on MS coco and segmentation results on ade20k, which the current work lacks.

**Questions:**

Please consider the Weaknesses, particularly point 2. It is important to have results for detection and segmentation to ensure the backbone is learning features that are rich enough for dense prediction tasks and not just over-fitting to the image classification task on imagenet.

**Limitations:**

There is some discussion on limitations in section 6. However, the authors should include additional limitations regarding their method, particularly about the fact that it requires additional annotations that may not be available across all datasets.

---

> ### Author Rebuttal · Authors · 2024-08-07
>
> Thank you for your thorough review and comments. We are pleased to hear that you find the motivation of this paper clear and that it holds significant importance for the vision community. We are also glad that you consider the correlation analysis proposed in the paper to be novel and detailed, and that the writing is clear and easy to understand. Below are our responses to each of your comments.
>
> ---
>
> > **Q1**: The method is limited by the required annotations (for example foreground annotation) which limits its usefulness across other classification datasets and other tasks.
> >
>
> **A1**: We fully understand your concerns. While it is true that not all datasets or tasks provide foreground annotations, it is important to note that the amount of foreground annotation required for external state defect repair is actually quite minimal. In our experiments, ImageNet-S includes foreground annotations for 9,190 images, with 10 images per class from 919 classes in the ImageNet-1K training set, while approximately 1,200,000 images in the ImageNet-1K training set are not annotated. This means that the amount of required foreground annotation represents a very small fraction of the total (lines 254 to 256 of the paper).
>
> Additionally, our exploratory experiments show that selectively annotating misclassified images in the training set is more effective than random annotation. Therefore, in practical scenarios, manually annotating approximately 10 challenging samples per class can sufficiently enable external state defect repair for Mamba. Furthermore, for internal state defect repair, the proposed method does not require any additional annotations (lines 242 to 235 of the paper).
>
> In summary, the proposed method is highly practical and applicable to a wide range of classification datasets and tasks.
>
> ---
>
> > **Q2**: The paper does not evaluate on any detection/segmentation benchmarks. This is required to understand whether the features learned by the backbone are rich enough to generalize to these more complex tasks. All recent vision backbones, including the various mamba-based ones, report detection results on MS coco and segmentation results on ade20k, which the current work lacks.
> >
>
> **A2**: Thank you for your valuable feedback. We appreciate your suggestion to evaluate the generalization of the learned features on detection and segmentation benchmarks.
>
> It is important to clarify that the goal of this work is not to develop a general-purpose backbone. Instead, one of our primary motivations is to investigate the internal mechanisms of the Mamba model from a post-hoc perspective, identify operational flaws in specific tasks, and repair them. This approach focuses on optimizing the model based on interpretability (lines 109 to 119 of the paper). Similar to many interpretability studies [1,2], we chose to validate our methods using widely adopted classification tasks. The results demonstrate that the proposed method effectively identifies and repairs defects in the Mamba model, thereby improving model accuracy.
>
> However, we acknowledge that adapting post-hoc interpretability methods to more complex tasks like detection and segmentation presents challenges. Most interpretability research has not yet been validated on detection/segmentation tasks, but this does not imply that our method lacks applicability in these areas. Indeed, the paradigm of our method is broadly applicable to Mamba-like architectures and can analyze defects from both external and internal state perspectives, regardless of the specific task.
>
> For detection and segmentation tasks, the main challenge is constructing correlations between external/internal states and prediction results. In classification tasks, these correlations are established using the predicted probabilities and gradients between the states and the true classes (Eq. (7) and Eq. (9) in the paper). For tasks like segmentation, where predictions involve probabilities for each pixel in the image, it is necessary to associate external/internal states with all pixels. One potential approach is to aggregate the predicted probabilities for different segmented regions from the ground truth, and calculate the gradients between the total prediction probabilities for each region and the states to construct these correlations. This can be used to analyze and constrain the defects in external/internal state correlations when segmentation results are poor (similar to Eq. (8) and Eq. (9) in the paper). Similarly, for detection tasks, it would be necessary to establish correlations between external/internal states and both the detected categories and spatial coordinates, analyzing these correlations to identify and repair defects.
>
> In summary, while the proposed post-hoc defect identification and repair paradigm for Mamba-like architectures is applicable to complex detection and segmentation tasks, specific modifications are needed due to the unique nature of task outputs. This remains a focus of our current and future work. We appreciate the reviewer's concerns regarding the validation on classification tasks and hope this response addresses your concerns.
>
> [1] Ali, Ameen, Itamar Zimerman, and Lior Wolf. "The hidden attention of mamba models." *arXiv preprint arXiv:2403.01590* (2024).
>
> [2] Jafari, Farnoush Rezaei, et al. "MambaLRP: Explaining Selective State Space Sequence Models." *arXiv preprint arXiv:2406.07592* (2024).

---

> ### Author Response · Authors · 2024-08-07
> **Rebuttal by Authors [Q3&Q4]**
>
> > **Q3**: Please consider the Weaknesses, particularly point 2. It is important to have results for detection and segmentation to ensure the backbone is learning features that are rich enough for dense prediction tasks and not just over-fitting to the image classification task on imagenet.
> >
>
> **A3**: Thank you for your comment. We sincerely hope that the previous response addresses your concerns. It is important to reiterate that flaw identification and repair are post-hoc optimization methods applied to a trained model. Like many studies exploring internal mechanisms of models, the patterns discovered are specific to that model. Consequently, the backbone obtained from flaw identification and repair on a classification task with Mamba may not be directly applicable to detection or segmentation tasks. However, as mentioned earlier, the proposed flaw identification and repair methods are adaptable to detection and segmentation tasks.
>
> ---
>
> > **Q4**: Limitations: There is some discussion on limitations in section 6. However, the authors should include additional limitations regarding their method, particularly about the fact that it requires additional annotations that may not be available across all datasets.
> >
>
> **A4**: Thank you for your valuable feedback. We agree that discussing additional limitations would make the paper more comprehensive. We have added the following content to the paper to better reflect the limitations of our method:
>
> “In the context of external state defect repair, while our method performs well with a small amount of annotations, it is undeniable that, with limited annotation resources, even manual labeling of a few samples may affect the usability of the proposed method. Similar to internal state defect repair, exploring defect repair methods that do not require additional annotations is a direction we will continue to investigate.”
>
> We hope that this response addresses your concerns.

---

> > ### Author Response · Authors · 2024-08-12
> >
> > Thank you for your valuable feedback and comments. We have provided our responses to your concerns above. Should you have any further questions or require additional clarifications, we would be more than happy to discuss them with you.

---

> > > ### Comment · Reviewer_3tmX · 2024-08-13
> > >
> > > Dear Authors,
> > >
> > > Thank you for your response. My concerns have been addressed partially. However, the main point for the request for detection/segmentation is to understand whether features learned by your method are truly rich enough, or are just over-fitting to the classification task. Which is why it is common practice in literature to use the ImageNet pretrained backbone in detection and segmentation. I believe this experiment is fundamental to the analysis important, and therefore I will be keeping my rating.

---

> > > > ### Author Response · Authors · 2024-08-14
> > > >
> > > > Dear Reviewer 3tmX and Reviewer nvv2,
> > > >
> > > > Thank you both for your continued engagement and thoughtful feedback.
> > > >
> > > > We appreciate  Reviewer 3tmX’s perspective on the importance of evaluating the generalizability of features learned by our method through detection and segmentation tasks. While we understand the significance of such experiments, we would like to reiterate that the primary focus of our study is to analyze the internal mechanisms of the Mamba model from a post-hoc interpretability standpoint, specifically within the context of classification tasks.
> > > >
> > > > As highlighted by Reviewer nvv2, the primary objective of our work is not to develop or evaluate a general-purpose backbone, but rather to perform a detailed post-hoc analysis of the Mamba model. Our focus is on understanding and enhancing specific aspects of the model's internal mechanisms, particularly in the context of classification tasks, which is a widely accepted approach in related studies. Our choice to validate on classification tasks aligns with the goals of the study, as it allows us to systematically identify and repair flaws within the model, thereby improving its accuracy.
> > > >
> > > > That said, we fully acknowledge that extending our evaluation to include detection and segmentation tasks is an important future direction. Furthermore, as mentioned in our response to your second question (Q2), the implementation of post-hoc analysis on the Mamba model in classification tasks differs somewhat from that in detection or segmentation tasks. We believe that our internal mechanism analysis and optimization paradigm hold potential for application in more complex tasks as well. We are actively working on this as part of our ongoing research.
> > > >
> > > > Once again, we sincerely thank you both for your constructive feedback, and we hope that this clarification helps to contextualize our approach within the intended scope of the paper.
> > > >
> > > > Best regards,
> > > >
> > > > Authors

---

> ### Comment · Reviewer_nvv2 · 2024-08-13
>
> Dear Reviewer 3tmx,
>
> Please allow me to join your discussion. I agree with the authors that this paper may not need to provide comprehensive experimental results in downstream tasks because this manuscript is more like a machine learning analysis paper. From my side, I think it is acceptable to only provide the results in classification tasks and analyze the inner mechanisms of the model architecture using this basic task. This is the style many machine learning scientists follow in their research.

---

> > ### Author Response · Authors · 2024-08-14
> >
> > Dear Reviewer 3tmX and Reviewer nvv2,
> >
> > Thank you both for your continued engagement and thoughtful feedback.
> >
> > We appreciate  Reviewer 3tmX’s perspective on the importance of evaluating the generalizability of features learned by our method through detection and segmentation tasks. While we understand the significance of such experiments, we would like to reiterate that the primary focus of our study is to analyze the internal mechanisms of the Mamba model from a post-hoc interpretability standpoint, specifically within the context of classification tasks.
> >
> > As highlighted by Reviewer nvv2, the primary objective of our work is not to develop or evaluate a general-purpose backbone, but rather to perform a detailed post-hoc analysis of the Mamba model. Our focus is on understanding and enhancing specific aspects of the model's internal mechanisms, particularly in the context of classification tasks, which is a widely accepted approach in related studies. Our choice to validate on classification tasks aligns with the goals of the study, as it allows us to systematically identify and repair flaws within the model, thereby improving its accuracy.
> >
> > That said, we fully acknowledge that extending our evaluation to include detection and segmentation tasks is an important future direction. Furthermore, as mentioned in our response to your second question (Q2), the implementation of post-hoc analysis on the Mamba model in classification tasks differs somewhat from that in detection or segmentation tasks. We believe that our internal mechanism analysis and optimization paradigm hold potential for application in more complex tasks as well. We are actively working on this as part of our ongoing research.
> >
> > Once again, we sincerely thank you both for your constructive feedback, and we hope that this clarification helps to contextualize our approach within the intended scope of the paper.
> >
> > Best regards,
> >
> > Authors

---

### Official Review · Reviewer_nvv2 · 2024-07-13

**Soundness:** 3
**Presentation:** 3
**Contribution:** 3
**Rating:** 6
**Confidence:** 4

**Summary:**

This paper analysis Mamba model from a post-perspective. It introduce a state correlation analysis method to establish the correlation between hidden states and predicted results, and analysis the external state flaws and internel state flaws.  Furthermore, this manuscript propose repair method to handle these flaws. Extensive experiments show its advantanges.

**Strengths:**

1. This paper is well written and easy to follow.

2. The topic is interesting and novel, which propose a new analysis perspective for the recent model Mamba.

3. Since the Mamba model is very popular recently, analysis this model is well motivate.

4. The experiment results are abundant and solid, as well as the detailed experiment setting.

**Weaknesses:**

1. One importent point is the classification task is not necessary only rely on the foreground pixels. Background regions may provide useful information some times. So the proposed external flaws may not be totally reasonable.

2. Only classification on ImageNet results are provides. It it suggented to provide more diverse experiments to full evaluation the proposed repair method.

3. Minors:

(1) Not all symbols are explained from Eq.(1)~(6).

(2) Missing punctuation (comma or full stop) in Eq.(1)~(6).

(3) In Fig.5, there are only (a)(b)(c)(d) in subfigures, while authors use (1)(2)(3)(4) in the caption.

**Questions:**

see weakness

**Limitations:**

see weakness

---

> ### Author Rebuttal · Authors · 2024-08-07
>
> Thank you for your thorough review and comments. We are pleased that you find our research topic both interesting and novel, and that you believe the analysis of the Mamba model is well-motivated. We are also gratified that the experimental results have met your approval. Below are our responses to each of your comments.
>
> ---
>
> > **Q1**: One important point is the classification task is not necessary only rely on the foreground pixels. Background regions may provide useful information some times. So the proposed external flaws may not be totally reasonable.
> >
>
> **A1**: Thank you for your valuable feedback. We agree that background regions can sometimes provide useful information. However, it is important to clarify that external flaws are identified through the analysis of external state correlation scores (Section 3.1). These scores reflect the interpretability of external correlations, specifically the degree to which the predicted results are associated with the foreground object regions (lines 146 to 159 of the paper). By comparing external state correlation scores between simple and difficult samples, we observed a significant drop in the scores for difficult samples (lines 196 to 203). This indicates that external flaws do not imply a complete disregard for background information, but rather that the Mamba model tends to focus more on background information in difficult samples, potentially neglecting important foreground details.
>
> Furthermore, during the defect repair process, we only constrained a small number of samples to focus on the foreground (lines 254 to 256 of the paper). This means that most samples are free to learn useful information from the background if it is beneficial, and our results demonstrate that this approach indeed improves the performance of the Mamba model.
>
> ---
>
> > **Q2**: Only classification on ImageNet results are provides. It it suggested to provide more diverse experiments to full evaluation the proposed repair method.
> >
>
> **A2**: Thank you for your valuable feedback. ImageNet is a widely used large-scale dataset for classification experiments. To thoroughly evaluate the proposed repair method within the limited time available for the rebuttal, we have also tested the ViM model on the smaller CIFAR-10 dataset. The results of these experiments are shown bellow:
>
> CIFAR-10 | Base: 84.25 | Flaw Repair: 85.77,
>
> where “Base” refers to the original model, and “Flaw Repair” refers to the proposed defect repair method.
>
> As shown above, the proposed defect repair method is also applicable to smaller datasets. For instance, the accuracy of the ViM model on CIFAR-10 improved by 1.50% after applying defect repair. It is worth noting that due to the small image size of CIFAR-10, the repair method in this experiment only addresses internal state defects.
>
> Additionally, to further validate the effectiveness of the proposed repair method, we conducted comparative experiments on the external and internal state correlations in the Mamba model before and after defect repair. These comparisons are illustrated in Figure 5 and described in lines 268 to 275 of the paper. Figure 5 shows that prior to defect repair, both external and internal state correlations of the Mamba model had low scores, indicating that the model associated incorrect regions during predictions. After defect repair, the scores for both external and internal state correlations improved, demonstrating that the model associated more accurate regions during predictions.
>
> In summary, the proposed repair method has broad applicability. We hope this response addresses your concerns, and we will update the paper with additional experiments to further validate the proposed repair method.
>
> ---
>
> > **Q3**: Minors: (1) Not all symbols are explained from Eq.(1)~(6).
> >
>
> **A3**: Thank you for your careful review and observation. We indeed overlooked providing explanations for some symbols. We have now added the following explanations after Eq. (1)~(6):
>
> “Here, SiLU$(.)$, causal-Conv1D$(.)$, and selective-SSM$(.)$ denote the activation function, the casual 1D convolution, and the selective state model, respectively. $W_x^{(\ell)}$, $W_z^{(\ell)}$, and $W_y^{(\ell)}$ are the projection matrices for the linear operations in the Mamba block, while $x$, $z$, $c$, $s$, and $y$ represent the intermediate states within the Mamba block.”
>
> Thank you once again for pointing out this detail.
>
> ---
>
> > **Q4**: *Minors: (2) Missing punctuation (comma or full stop) in Eq.(1)~(6).*
> >
>
> **A4**: Thank you for your meticulous review and observation. We have added the missing commas and full stops in *Eq.(1)~(6)* and have rechecked the entire manuscript.
>
> ---
>
> > **Q5**: Minors: (3) In Fig.5, there are only (a)(b)(c)(d) in subfigures, while authors use (1)(2)(3)(4) in the caption.
> >
>
> **A5**: We apologize for the typographical errors. We have corrected these discrepancies in the manuscript and have carefully reviewed the entire paper to ensure consistency.

---

> > ### Author Response · Authors · 2024-08-12
> >
> > Thank you for your valuable feedback and comments. We have provided our responses to your concerns above. Should you have any further questions or require additional clarifications, we would be more than happy to discuss them with you.

---

> > > ### Comment · Reviewer_nvv2 · 2024-08-12
> > >
> > > The rebuttal addresses all my concerns. I decide to increase my rating from borderline accept to weak accept. The authors should add the points from rebuttal into the final version if this manuscript is finally accepted.

---

> ### Author Response · Authors · 2024-08-14
>
> Thank you for your thorough review and valuable suggestions. We are pleased that our responses have addressed your concerns. We will incorporate these points into the final version of the manuscript. Once again, we appreciate your contribution to improving this work.

---

### Official Review · Reviewer_4vzy · 2024-07-13

**Soundness:** 3
**Presentation:** 3
**Contribution:** 3
**Rating:** 7
**Confidence:** 5

**Summary:**

1. Addressing the main issue in the existing model:
   Mamba, despite its success in long sequence tasks, faces mixed opinions and challenges in visual tasks due to inherent flaws and suboptimal performance. Understanding these flaws and optimizing Mamba's performance in the visual domain are critical research questions.

2. Proposed solution for the issue:
   The paper proposes Vision Mamba Mender, a systematic approach to enhance Mamba's performance in visual recognition tasks. This approach involves predictive correlation analysis of Mamba's hidden states, both internally and externally, to identify and address flaws. Tailored repair methods are then applied to optimize model performance.

3. How did the author evaluate the proposed solution on various standard dataset:
   The efficacy of Vision Mamba Mender is validated through extensive experiments on prevalent Mamba architectures. These experiments demonstrate significant improvements in model performance across various standard datasets, showcasing the practical impact and effectiveness of the proposed methods. The algorithm code is also provided for transparency and reproducibility.

**Strengths:**

1. Innovative Post-Hoc Optimization: Vision Mamba Mender introduces a novel approach to optimize Mamba models post-training, focusing on identifying and rectifying operational flaws rather than predefining architectures. This method is innovative in its systematic approach to improving model performance.

2. Comprehensive State Analysis: The paper introduces a detailed state correlation analysis method that evaluates Mamba's hidden states from both external and internal perspectives. This comprehensive analysis helps in pinpointing specific flaws that affect prediction accuracy in visual recognition tasks.

3. Tailored Repair Methods: Tailored repair methods are proposed for both external and internal state flaws identified through the analysis. By imposing constraints on state correlations within Mamba modules, these methods effectively enhance the model's ability to make accurate predictions.

4. Applicability to Existing Architectures: Vision Mamba Mender is designed to be applicable across various state-of-the-art Vision Mamba architectures. This versatility ensures that the optimization approach can be seamlessly integrated into different implementations without significant modifications.

5. Experimental Validation: Extensive experiments validate the effectiveness of Vision Mamba Mender across different benchmarks and datasets. The approach consistently demonstrates improvements in model accuracy, showcasing its practical utility in real-world applications.

6. Transparency and Reproducibility: The availability of algorithm code in the supplementary material and commitment to making it publicly accessible enhance the transparency and reproducibility of the research findings. This openness facilitates further validation and adoption of the proposed methods by the research community.

**Weaknesses:**

1. Limitation in Methodology: The paper mentions the introduction of state correlation analysis and repair methods for Mamba, but it lacks clarity on the specific algorithms or mathematical formulations used. This could hinder reproducibility and transparency, as other researchers may find it challenging to implement or verify the proposed methods without detailed methodology descriptions.

2. Assumptions on Flaws and Repair Methods: The paper suggests that Vision Mamba Mender identifies and rectifies anomalies in Mamba's mechanisms, focusing on external and internal state correlations. However, without empirical evidence or case studies illustrating the nature of these flaws across different datasets and scenarios, the validity and generality of these assumptions remain unclear.

**Questions:**

Please refer to the Weaknesses block

**Limitations:**

Please refer to the Weaknesses block

---

> ### Author Rebuttal · Authors · 2024-08-07
>
> Thank you for your comments and the positive feedback. We have carefully reviewed each of your comments. Although some comments may not fully align with our research objectives, we are nonetheless very appreciative of your feedback. Below are our responses to each of your comments.
>
> ---
>
> > **Q1**: The paper mentions the introduction of state correlation analysis and repair methods for Mamba, but it lacks clarity on the specific algorithms or mathematical formulations used. This could hinder reproducibility and transparency, as other researchers may find it challenging to implement or verify the proposed methods without detailed methodology descriptions.
> >
>
> **A1**: Thank you for your valuable comments. We understand your concerns. In response, we have detailed the methods proposed for the novel Mamba model in the visual domain to ensure clarity and reproducibility.
>
> In Section 3 of the paper, we thoroughly describe the computational process of the Mamba model using numerous formulas. Subsequently, in Section 4, we provide a detailed mathematical formulation and definitions for the algorithms used to identify defects in the Mamba model, including how to establish external/internal state correlations and how to define correlation scores. Finally, in Section 5, we offer a comprehensive description, using mathematical formulas, of the algorithms employed to repair identified defects in the Mamba model, including specific methods for addressing external/internal state defects.
>
> Additionally, we have included detailed experimental setups in Appendix D. We hope these measures will assist readers in implementing our proposed algorithms effectively and address your concerns.
>
> ---
>
> > **Q2**: The paper suggests that Vision Mamba Mender identifies and rectifies anomalies in Mamba's mechanisms, focusing on external and internal state correlations. However, without empirical evidence or case studies illustrating the nature of these flaws across different datasets and scenarios, the validity and generality of these assumptions remain unclear.
> >
>
> **A2**: Thank you for your comments. We fully understand your concerns.
>
> The approach proposed in our paper, which is based on external and internal state correlations, aims to explore the mechanisms of the Mamba model from these two perspectives. We have demonstrated the presence of external and internal state defects in the Mamba model through a comparison of state correlation scores for simple and challenging samples. Extensive experiments and analyses are detailed in Section 3.3 of the paper, as well as in Appendices E and F.
>
> Furthermore, in the field of computer vision, other researchers have also attempted to understand the workings of the Mamba model, similar to our approach. For example, Ali et al. [1] have explored the mechanisms of Mamba in image recognition tasks and their findings reveal some external state anomalies. However, their work primarily focuses on external state correlations and lacks an analysis of internal state correlations. Additionally, their methods are applicable only to state space models and cannot be used for other computations within the Mamba block (as discussed in lines 113 to 119 of our paper).
>
> In summary, this paper is the first to propose a post-hoc optimization approach for the Mamba model by analyzing both external and internal state correlations to identify and repair defects. This not only improves model performance but also aids readers in understanding the mechanisms of the novel Mamba model. We appreciate your feedback and hope the above response addresses your concerns.
>
> [1] Ali, Ameen, Itamar Zimerman, and Lior Wolf. "The hidden attention of mamba models." *arXiv preprint arXiv:2403.01590* (2024).

---

> > ### Author Response · Authors · 2024-08-12
> >
> > Thank you for your valuable feedback and comments. We have provided our responses to your concerns above. Should you have any further questions or require additional clarifications, we would be more than happy to discuss them with you.

---

> > > ### Comment · Reviewer_4vzy · 2024-08-12
> > > **Vision Mamba Mender**
> > >
> > > Thank you for your response. Most of my concerns have been addressed in the rebuttal.

---

> ### Author Response · Authors · 2024-08-14
>
> Thank you for your diligent review and comments. We are pleased that our responses have addressed your concerns.  We appreciate your support for this work.

---

### Author Rebuttal · Authors · 2024-08-07

Dear Reviewers,

Thank you for the time and effort you have invested in reviewing our paper. We are particularly grateful for your recognition of the novelty and originality of our work. We are also pleased that our approach to analyzing and optimizing the Mamba model from a post-hoc perspective has contributed to a better understanding of this new model and its performance improvements.

In addressing each of your comments and concerns, we believe that the paper has been significantly improved. We have individually responded to each comment and collected the following common issues raised by reviewers. If you find that our replies address your concerns, we would be grateful if you could consider raising the score.  Should you have any further questions, we are more than happy to engage in additional discussions.

---

>  Reviewer JDBm expressed concerns about the details of the flaw identification method for the Mamba model and the limitations of analyzing and enhancing the Mamba model solely from a post-hoc perspective.
>

In response to these concerns, we have provided a detailed analysis and explanation of the defect identification methodology in our responses to the reviewer’s comments. Additionally, we have highlighted the significant advantages and importance of analyzing and enhancing the Mamba model from a post-hoc perspective compared to other approaches. We believe these responses address the reviewer’s concerns effectively.

---

> Reviewers nvv2 and 3tmX both raised concerns regarding the relevance of foreground information in the identification and repair of external state flaws. Reviewer nvv2 pointed out that classification tasks do not solely rely on foreground information, as background regions can sometimes provide useful data. Reviewer 3tmX noted that the requirement for foreground annotations limits the applicability of the proposed method to other classification datasets and tasks.
>

Thank you for the valuable feedback. We apologize for any confusion caused. We have provided detailed responses to each reviewer’s comments below and believe these responses address their concerns. It is important to emphasize again that the proposed method for identifying and repairing external state flaws does not restrict the Mamba model from learning background information. Furthermore, the amount of foreground annotation required for repairing external state flaws is minimal, making the proposed method both reasonable and practical.

---

> Reviewer 3tmX expressed concerns about the lack of evaluation of backbone generalization on more complex tasks.
>

We understand the reviewer’s concerns. To address this, we have provided detailed responses regarding why the proposed method was validated only on classification tasks, how the method can generalize to more complex tasks, and how it can be adapted for such tasks. In summary, the goal of this work is not to develop a universally applicable backbone for various complex tasks. Instead, one of the motivations for this work is to analyze the internal workings of the Mamba model from a post-hoc perspective, identify specific operational flaws in certain tasks, and repair them. The paradigm of identifying and repairing flaws from both external and internal state perspectives is applicable and valuable.

---

> Reviewers 4vzy, nvv2, and JDBm pointed out issues with the explanations of formulas and some typographical errors in the paper.
>

We appreciate the reviewers' attention to these details. We have corrected these issues in the paper and thoroughly checked the entire text to ensure accuracy.

---

Besides the issues mentioned above, more detailed responses to the reviewers' comments can be found below each reviewer's specific feedback. We sincerely thank all the reviewers for their diligent review and valuable suggestions, which have greatly improved the paper.

---

### Decision · Program_Chairs · 2024-09-25

**Decision:**

Accept (poster)

**Comment:**

This paper proposes to explore the internal learning mechanism of mamba, identifying mamba flows and thus optimizing the design. It received 3 positive reviews and 1 negative review initially. The raised concerns include method limitations, flaw assumption validation, only classification results provided. In the rebuttal phase, the authors have addressed several issues by responding in details. Finally, the remaining issue is not experimenting object detection / segmentation scenarios. The AC has checked all files, and agree to reviewers that the current form shall raise reporting. The authors are suggested to provide det/seg results in the camera-ready version. Congratulations.